# Pediatric Healthcare Utilization in a Large Cohort of Refugee Children Entering Western Europe During the Migrant Crisis

**DOI:** 10.3390/ijerph16224415

**Published:** 2019-11-11

**Authors:** Christine Happle, Christian Dopfer, Diana Ernst, Evelyn Kleinert, Annabelle Vakilzadeh, Susanne Hellms, Iro Evlampidou, Nele Hillermann, Reinhold E. Schmidt, Georg M. N. Behrens, Frank Müller, Martin Wetzke, Alexandra Jablonka

**Affiliations:** 1Department of Pediatric Pneumology, Allergology, and Neonatology, Hannover Medical School, 30625 Hannover, Germany; happle.christine@mh-hannover.de (C.H.); dopfer.christian@mh-hannover.de (C.D.); wetzke.martin@mh-hannover.de (M.W.); 2German Center for Lung Research, Biomedical Research in End Stage and Obstructive Lung Disease/BREATH, 30625 Hannover, Germany; 3Department of Clinical Immunology and Rheumatology, Hannover Medical School, 30625 Hannover, Germany; ernst.diana@mh-hannover.de (D.E.); immunologie@mh-hannover.de (R.E.S.); behrens.georg@mh-hannover.de (G.M.N.B.); 4Department of General Practice, University Medical Center Göttingen, 37073 Göttingen, Germany; evelyn.kleinert@med.uni-goettingen.de (E.K.); nelehillermann@web.de (N.H.); frank.mueller@med.uni-goettingen.de (F.M.); 5Hannover Medical School, 30625 Hannover, Germany; Vakilzadeh.Annabelle@mh-hannover.de; 6Institute for Diagnostic and Interventional Radiology, Hannover Medical School, 30625 Hannover, Germany; hellms.susanne@mh-hannover.de; 7MediPIET, 5 Madrid, Spain; Iro.Evlampidou@gmail.com; 8German Center for Infection Research (DZIF), partner site Hannover-Braunschweig, 30625 Hannover, Germany

**Keywords:** children, pediatrics, healthcare, migration, refugee, primary care, adolescents, migrant, health care

## Abstract

Background: Currently, half of the population displaced worldwide is children and adolescents. Little is known on healthcare demand in underage migrants. Materials and Methods: We analyzed healthcare utilization in n = 1.411 children and adolescents living in a large German refugee reception in 2015-2016. Results: The mean age of all included refugees was 9 years (60.8% male). The majority came from the eastern Mediterranean region. During a mean camp inhabitance of 34 days, 57.6% needed primary healthcare, with a significant inverse correlation of healthcare seeking frequency with age and duration of camp inhabitance. Infants and unaccompanied minors displayed particular high demands for medical help. Discussion: Our analysis showed that pediatric primary healthcare in pediatric and adolescent refugees are most sought during the first period upon arrival with particular demand in infants, toddlers, and unaccompanied minors. Based on this data, future care taking strategies should be adapted accordingly.

## 1. Introduction

According to the United Nations Refugee Agency (UNHCR), roughly half (52%) of the population currently displaced worldwide is less than 18 years old [1]. Due to ongoing political crises and war with breakdown of the local medical system, millions of children and adolescents are constantly exposed to psychological and physical trauma and inadequate healthcare [1,2]. Routine medical data accumulated by Doctors Without Borders in northern Syria in 2013–2016 showed that underage patients were particularly prone to communicable diseases (one-fifth of medical care consultations in children below the age of 5 years were due to infectious diseases) and highlighted the impact of poorly treated chronic diseases in this population [3]. A plethora of further analyses showed that refugee children are at particular risk for diseases such as depression, infections, or poorly managed chronic diseases [4,5,6,7]. We and others previously identified children arriving in Europe during the current crisis to be particularly vulnerable to physical trauma and infectious diseases during their flight [8,9,10,11,12,13]. Besides psychological and physical trauma during displacement, malnutrition, poor hygiene, and overcrowded living conditions further add to the particular risk of refugee children for increased morbidity [1,14].

Humanitarian emergencies worldwide are currently posing new challenges to receiving healthcare systems in many countries. During the recent crisis in Europe, large numbers of potential patients arrived rather unexpectedly and needed immediate medical care [15]. In such instances, standard operating procedures were oftentimes not applicable, given the extent of the situations.

We previously showed that health care utilization is high in freshly arriving immigrants in western Europe during the current crisis, especially directly upon arrival at a primary refugee reception center [16]. Another study on primary medical care in newly arriving immigrants in Greece showed a similarly high frequency of healthcare-seeking behavior, especially due to infectious and dental pathologies, but also because of psychological diagnoses [13]. In Turkish regions close to the Syrian border, refugee children were significantly more likely to visit local emergency departments than local children were [17]. Another study by Watts et al. showed that refugee children not only displayed a comparably high disease burden when arriving at a country with a high economic standard, but that these diseases also led to significant healthcare demand during the first year after resettlement [18]. However, in spite of the high proportion of migrating children and refugees, data on their particular medical needs during the current crisis were scarce.

To improve this situation and prepare for future immigration scenarios, a thorough analysis of healthcare utilization during times of significant migration is central [19]. Accordingly, we aimed to identify healthcare-utilization profiles and socio-demographic characteristics of migrating children and adolescents during the peak of immigration in 2015.

## 2. Methods

### 2.1. Study Population

There were n = 3104 migrants housed at a temporary reception site in Celle, northern Germany, from its opening on 4 September 2015 to closing on 4 July 2016. This study analyzed all n = 1423 children and adolescents admitted to this emergency refugee shelter. The median duration of camp stay was 34 days. All children and adolescents who entered the camp from opening to closing and all medical data were included into the analysis. All camp residents had been allocated to the shelter without preselection. Allocation was based on time point of arrival to Germany through an official German allocation key (Königsteiner Schlüssel) and all refugee shelter inhabitants were registered upon arrival and departure from the camp. In case inhabitants left the reception site without official notice, the last contact date with camp staff (e.g., for medical service, food service, transportation) was assumed as departure day. Parts of the cohort analyzed here have been previously described [15,16,17]. This paper uses the somewhat disputable term “refugees” for subjects within this cohort. Almost all camp inhabitants were asylum seekers in the beginning; most of them became refugees according to their legal status in Germany during their stay at the camp, and all of the migrants left their home countries to temporarily or constantly live in a peaceful country, hence the simplification “refugee” appears to be justified in this context.

### 2.2. Data Collection

All data were collected during routine clinical care. Information about age, gender, country of origin, healthcare utilization, language knowledge, and attendance dates of camp residents were extracted from an electronic database. Prior to scientific analysis, all data were fully anonymized. At the refugee shelter, a 24 h/7 days full-time medical service was available to all camp residents with free primary medical service. Health professionals volunteered to provide medical care. Paramedics were located at the improvised onsite outpatient clinic full-time, and medical doctors were available daily for consultations until December 2015. From January 2016, medical doctors were present at least every-other day and could be consulted daily in case of medical emergencies. At the onsite ward, basic laboratory tests, ultrasound, and electrocardiographic recordings could be performed. For specific complaints, planned visits of pediatricians, psychiatrists, surgeons, or other specialized professionals were scheduled, and referral to further specialists or hospital care was arranged whenever needed. Every onsite healthcare utilization was documented in an electronic database. Medical primary check-ups as required by German asylum law were provided to all refugees in a nearby hospital. These check-ups are not part of this data analysis.

### 2.3. Operational Definitions

This analysis includes pediatric patients, consisting of children and adolescents of up to 18 years of age who sought help at least once at the onsite medical facilities. Age upon arrival was used for the analysis, and the following age definitions were used: Infants < 1year, toddlers 1–3 years, preschool 4–5 years, school children 6–12 years, young teenager 13–15, old teenager 16–18 years of age. Countries of origin were self-reported by the refugees. The following countries were included in the regions. Africa: Ethiopia, Burundi, Ivory Coast, Eritrea, Gabon, Gambia, Guinea, Cameroon, Liberia, Mali, Mozambique, Namibia, Niger, Nigeria, Rwanda, Zimbabwe, Somalia, Sudan, and Togo. Asia: Bangladesh, Philippines, and Vietnam. Europe: Albania, Bosnia, Kosovo, Macedonia, Republic of Moldavia, Montenegro, and Serbia. Commonwealth of Independent States (CIS) or states of the former Soviet Union: Armenia, Azerbaijan, Georgia, Kazakhstan, Russia, Chechnya, Ukraine, and Uzbekistan. Middle East: India, Nepal, Pakistan, Afghanistan, and Iran.

### 2.4. Statistical Analyses

Healthcare utilization was calculated as frequency of visits to the onsite medical services in relation to duration of personal camp inhabitance (consultations/day of camp inhabitance). Normal distribution of all variables was assessed, and associations between metric variables were assessed by linear regression. Depending on data distribution, group differences with categorical items were evaluated by Mann–Whitney U test, Pearson’s Chi-Sqare test, or one-way ANOVA with Kruskal–Wallis and Dunn´s correction. Correlation between dichotomous variables or metric variables was evaluated with Pearson’s correlation when the values were distributed normally. Otherwise Spearman’s correlation was used. P values below 0.05 were considered significant. IBM SPSS Statistics version 24.0 (Armonk, New York, NY, USA) and Graphpad Prism version 5.02 (San Diego, CA, USA) were used. For the analysis of reasons for medical visits as presented in Appendix A, medical records of n = 100 random visits of children were taken from the entire dataset employing SPSS, and International Statistical Classification of Diseases and Related Health Problems (ICD-10) coding was used to describe the patient complaints.

### 2.5. Ethics Compliance

The analyses presented here were approved by local authorities (Institutional Review Board of Hannover Medical School approval #3217-2016).

## 3. Results

### 3.1. Cohort Characteristics

Of n = 3104 refugees inhabiting the camp, n = 1423 (45.8%) were 18 years of age or younger. Age and gender distribution of the cohort are depicted in Figure 1A. The median age of these refugees was 9 years (interquartile range (IQR) 11), and 60.8% of them were male, and n = 20 (0.006%) of them were unaccompanied minors. Exact dates of camp entrance and exit were available for n = 1411 children or adolescent refugees (99.2%). The median duration of camp stay was 34 days (IQR 55). Most children and adolescents came from the eastern Mediterranean region (63.9%) or the Middle East (26.6%). Only a comparably small proportion came from Europe (3.4%), Africa (0.9%)/ North Africa (0.7%), Asia (0.5%), or states of the former Soviet Union (0.4%). In 3.4% of children and adolescents, the country of origin was not reported (Figure 1B). There were 41.9% who spoke Arabic languages, followed by 23.4% who spoke Persian languages (Figure 1C). Only 4.5% of the camp inhabitants below the age of 18 years spoke English, 0.3% French, and 0.2% German. In 11.2%, no language knowledge was recorded, i.e., in infants (Appendix A).

### 3.2. Healthcare Utilization and Patient Characteristics within the Analyzed Cohort

In total, 57.6% (n = 820) of all refugee children and adolescents presented at least once to the onsite medical ward (male 53.9%, female 63.4%). Overall, this was not significantly different to the healthcare-seeking behavior of adult inhabitants of the camp, which displayed an overall proportion of 59.2% of refugees seeking help at the onsite ward. Overall, 44.1% of medical visits occurred in pediatric patients. The demographic characteristics of the pediatric patients did not differ significantly from those of the overall pediatric cohort: The median age of pediatric patients presenting to the onsite medical ward was 7 years (IQR 10, male 8 years (IQR 10); female 6 years (IQR 9)), 61% of patients were male, and the majority of patients (61.3%) came from the eastern Mediterranean region.

Overall, a median visit frequency of 0.04 per child and day of camp residence was observed (IQR 0.1), and the median number of visits per patient was 5 (IQR 9).

However, specific demographic factors impacted the pediatric healthcare-utilization behavior in our cohort. When we compared the onsite medical ward visit frequency per day of camp inhabitance in male versus female underage patients within our cohort, we found a slightly but significantly higher rate of healthcare utilization in females (Figure 2A). When analyzing age-specific subgroups, we observed the highest frequency of healthcare utilization in the youngest age group: 74.1 % of infants presented to the medical center, compared to only 44.3% of the 16–18 year old refugees. Infants and toddlers presented significantly more often to the medical team compared to teenagers (median 0.0804 (average 0.1267) visits per person and day vs. 0.0000 (average 0.0363)). Similarly, school children sought significantly more medical help than younger and older teenagers (median 0.0370 (average 0.0598) visits per person and day vs. 0 (0.0335) vs. 0 (0.0383)) (Figure 2B). Accordingly, when analyzing patient-specific age and visit frequencies, an inverse correlation was observed (Figure 2C, Pearson’s correlation *r* = −0.26, *p* < 0.001).

Also, the personal duration of stay at the camp impacted healthcare utilization significantly.

Overall, 53.2% of all pediatric medical consultations occurred within the first week after camp entrance. In the first week upon arrival, 30.6% of all children and adolescents presented to the medical center compared to only 17.5% in week three after camp entrance. The mean number of visits of all underage refugees in the first week was 0.02/d compared to only 0.005/d in the third week and even lower healthcare utilization in subsequent weeks. Starting from week 8 of individual camp inhabitance, a fairly stable health care utilization of 0.002 visits per day was recorded (Figure 2D). When correlating personal duration of camp inhabitance values with visit frequency of medical visits per day, again a significant correlation occurred (data not shown, Spearman’s correlation *r* = 0.614, *p* < 0.001).

### 3.3. Influence of Origin, Unaccompanied Status, and Season on Healthcare Utilization within the Analyzed Cohort

With regard to regions of origin, we found the highest healthcare utilization in minor refugees from the eastern Mediterranean Region and Europe (Figure 3A). When focusing on the five most prevalent nationalities within our cohort, we found the highest rate of visit frequency in children and adolescents from Iran (mean 0.12), and a slightly but significantly higher rate of healthcare utilization in Iraqi compared to Syrian refugees (0.07 vs. 0.06, Figure 3B).

Within the analyzed cohort, only 20 children and adolescents reported to have arrived as “unaccompanied minors” in the sense that they had immigrated without family members or other personal caregivers. These unaccompanied minors were older than their accompanied counterparts (median: 15 years), predominantly of male gender (60.3%), and stemmed from Afghanistan (n = 9), Iraq (n = 6), and Syria (n = 5). In total, 18/20 (90%) unaccompanied minors used the provided medical service, and their healthcare utilization per day of residence at the reception center was significantly higher than that of children and adolescents migrating with a personal caregiver (Figure 3C, X² 8.707, *p* = 0.003).

Furthermore, we analyzed seasonal changes in healthcare utilization within our cohort. Here, we found healthcare utilization to be higher in autumn/winter, with a peak at calendar week 50 (Figure 3D).

Finally, we conducted a sample analysis to address the reasons for primary healthcare seeking in our cohort. For this, we took a random sample of n = 100 healthcare visits from the entire dataset and analyzed the complaints on the basis of patient’s medical record in these children and adolescents (47% male, mean age 6.5 yrs. (range 0–18 years)). The random visit sample did not significantly differ from total visits in terms of gender (*p* = 0.104), age (*p* = 0.329), region of origin (*p* = 0.941), and duration of stay (*p* = 0.877). In 89% of visits, a complaint was recorded. As shown in Appendix A, the main specific diagnosis groups based on ICD-10 recording included respiratory and infectious diseases (39.4444%), followed by diseases of the skin, ear, or gastrointestinal tract (Appendix A). When we further analyzed more specific diagnoses, complaints such as cough, respiratory disease, fever and infections, or gastroenteritis occurred as top diagnoses, again illustrating that these acute pathologies appeared to be the most prevalent complaints in our cohort (Appendix A).

## 4. Discussion

We here presented a comprehensive analysis of healthcare utilization in a large cohort of minor-aged refugees entering western Europe during the current crisis and show that medical attention is particularly needed during the first time upon arrival in Germany, and in particularly young children and unaccompanied minors.

Demographic characteristics of this cohort are representative for the current population immigrating towards Europe: Most children and adolescent refugees in this cohort came from the Middle East or eastern Mediterranean region, reflecting the currently high frequency of refugees from these regions, and the slight predominance of male refugees was mainly caused by the large proportion of male adolescents in the highest age groups (16–18 years) of our cohort, which is in line with current global migration statistics that show that a large proportion of people on the move currently are male adolescents and young adults [18,19,20]. Overall, 44% of all medical consultations at the onsite ward in the described reception center occurred in children and adolescents, which emphasizes the need for onsite pediatricians in humanitarian situations as the current crisis. Another finding in our analysis with direct implications for onsite medical care for refugees in the future was the fact that medical help was sought by children and adolescents most frequently in the days and weeks directly after arrival. We reported similar findings in a large cohort of refugees of all ages before [16] and suggest that this needs to be addressed when preparing supply and staff demand of an onsite medical ward at times of new refugee arrival. Also, the language statistics in our results may help to calculate the need for translators in similar settings in the future.

The healthcare-seeking rates in our cohort were considerably higher than those previously described by Hermans et al. for underage refugees at a camp in Lesbos, Greece [13]. In this study, all consecutive patients who visited doctors from the local “Boat Refugee Foundation” that provided onsite primary medical care at the refugee sites Camp Moria and Caritas hotel were analyzed to assess healthcare-seeking behavior during the current crisis. The authors found a mean of 0.01 visit per day in refugees at Lesbos as compared to 0.06 in our cohort arriving in Germany. The observed difference may be due to a different set-up of medical care provision and different recording strategies in the two studies. However, the healthcare-seeking rates within our cohort after week eight (mean visit of 0.008 per day) are somewhat comparable to those observed by Hermans et al. This rate might more adequately reflect the population in the Mori/Caritas hotel camps in Lesbos, where predominantly refugees with long-term camp residence are housed. Furthermore, our study included all medical encounters that took place in one central medical ward of our camp, while in other studies, such as in that by Hermans et al., multiple medical organizations provided health services to the population which may have led to incomplete or inconsistent recording of healthcare-seeking behavior in all children and adolescents. Furthermore, in other studies, barriers like vouchers, out-of-camp medical facilities, or language barriers may have hindered access to medical care [20,21]. The barriers in our setting were comparably low since onsite access to healthcare with interpreters was actively supported in our refugee housing situation.

The overall rate of healthcare seeking in children and adolescents was not significantly higher than that previously reported by us for refugees of all ages in a similar setting, which may be due to the high percentage of adolescents in our current analysis group who did not display an increased demand for medical care as compared to the previously analyzed migrants [16]. However, in our cohort, a significant inverse correlation of age and healthcare demand was found. This is in line with previous studies showing that that hospitalization rates for ambulatory care-sensitive conditions are higher in refugee children than in the general population with an odds ratio of 1.81 and the utilization of emergency use for ambulatory care-sensitive conditions was higher with an odds ratio of 4.93 [22]. A study on emergency healthcare-seeking behavior in Turkish regions close to the Syrian border showed that refugee children were significantly more likely to visit local emergency departments than local children [17]. Furthermore, it has been shown that refugee children do not only carry a comparably high disease burden upon arrival to their destination state, but that these diseases lead to significant healthcare demand during the first year after resettlement [18].

In our cohort, we observed an inverse correlation of age and healthcare utilization, as the youngest children (infants and toddlers) within our cohort displayed a significantly higher demand for medical help compared to school children and adolescents, and visits of preschool and younger school children to the onsite medical ward was significantly more compared to older teenagers. This finding is compatible with the previous reports on the higher demand for medical care in younger children [23]. Especially infants and toddlers are more susceptible to infectious diseases [24,25].

Also, the finding of a higher number of clinic visits in autumn and winter is in accordance with the high rate of communicable diseases in these seasons [26]. Although all housings were heated and all inhabitants of the refugee camp had regular access to sanitation, the crowdedness of the living situation rendered them particularly vulnerable to the fast-spreading of communicable diseases, particularly during cold season [8,9,12,13]. This is also reflected by the analysis of diagnoses in a sample of our cohort: Although we did not conduct a comprehensive analysis of all healthcare visits in our setting, our preliminary calculation showed that the main reasons for visiting the onsite medical team in our cohort were acute complaints such as infection, cough, fever, or other potentially infectious diseases. This observation, again, goes in line with that of our and other authors’ describing the increased infection susceptibility in young refugees and their increased healthcare demand due to infection-related pathologies or other acute complaints [9,27,28].

As the number of unaccompanied children and adolescent refugees within our cohort is extremely small (n = 20), we can only cautiously speculate that the observed differences in healthcare-seeking behavior in this subgroup reflects the demands of unaccompanied minors during the current crisis. We observed a higher healthcare-seeking rate in unaccompanied as compared to accompanied minors within our cohort. Although most of the unaccompanied minors in our cohort were adolescents of “almost adult” age, we can only speculate on the immense physical and psychological stress migration without a proper caretaker poses at this age. This may have led to a higher demand for medical help in this small subgroup of our cohort. This finding is in line with the high rate of physical and mental health issues in a study on health in unaccompanied minors by Marquardt et al.: In this study, 72% of unaccompanied pediatric or adolescent refugees with medical problems needed referral to a specialist and 75% received a prescription [29].

A limitation of our work is that particular subgroups within our cohort (such as unaccompanied minors or those stemming from particular regions or countries) are rather small and, as such, may not be representative for the large population currently on the move. Further research is necessary to evaluate whether healthcare demand is truly higher in underage migrants from the eastern Mediterranean region or Europe compared to those stemming from other regions. Another limitation of our work is that we could not analyze complaints in all visits but only provide a thorough analysis of complaints in a sample of children and adolescents of our cohort. Future studies will focus on these issues.

Taken together, our data illustrate the medical needs of refugee children and adolescents for primary care upon first arrival in a country with a high socioeconomic standard. Children and adolescents are especially vulnerable to physical and emotional trauma [30,31] and deserve particular medical attention and optimized care. As the UNHCR stated: Optimal care during the current refugee crisis demands a “multidimensional and comprehensive approach in public health and nutrition, and will require funding and donations of both technical support and commodities/funds beyond the normal programming needs” [27]. Our data illustrates the particular medical needs in this vulnerable population, and to adapt care-taking strategies accordingly.

## Figures and Tables

**Figure 1 ijerph-16-04415-f001:**
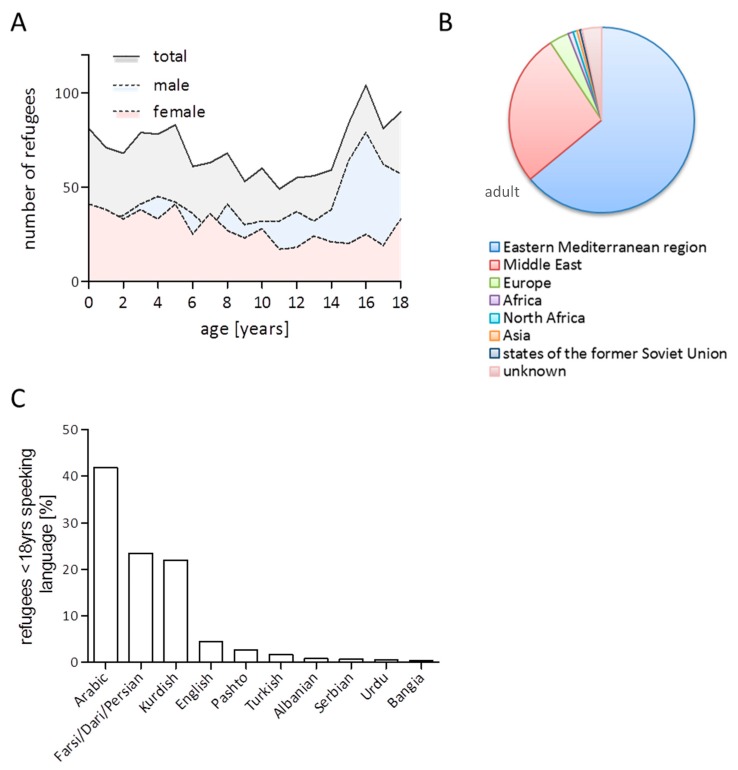
Age and gender distribution (**A**), regions of origin (**B**), and spoken language (**C**) within the analyzed cohort of refugees below the age of 18 years.

**Figure 2 ijerph-16-04415-f002:**
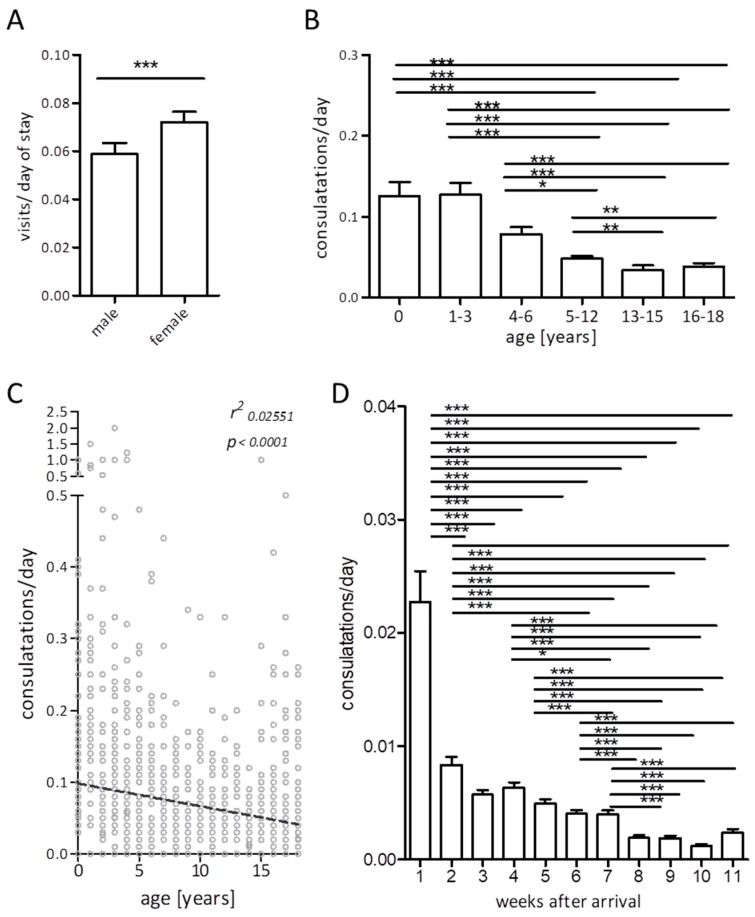
Factors impacting the visit frequencies normalized to day of stay at the refugee residence: Gender (**A**), age (**B**,**C**), duration of personal stay at the residence (bars display mean + Standard Error of Mean (**A**,**B**,**D**), linear regression in (**C**), * *p* ≤ 0.05, ** *p* ≤ 0.01 *** *p* ≤ 0.005).

**Figure 3 ijerph-16-04415-f003:**
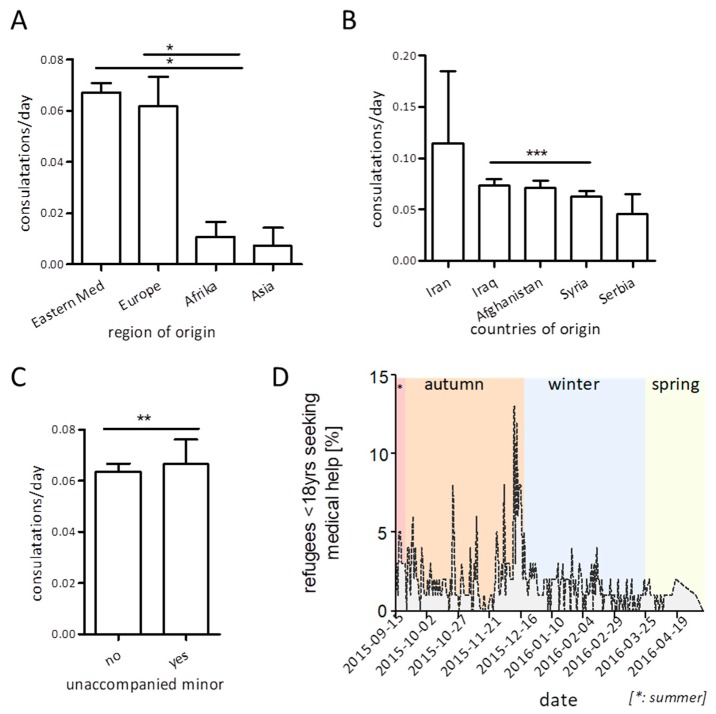
Visit frequencies normalized to day of stay at the refugee residence depending on regions (**A**) or country (**B**) of origin. Influence of family-accompany status (**C**) and seasonal changes (**D**) on healthcare utilization within the cohort (bars display mean +SEM (**A**,**B**,**D**), linear regression in (**C**), * *p* ≤ 0.05, ** *p* ≤ 0.01 *** *p* ≤ 0.005).

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
