# Peer review of "Pediatric Healthcare Utilization in a Large Cohort of Refugee Children Entering Western Europe During the Migrant Crisis"

_ijerph, 2019, doi:10.3390/ijerph16224415_

Round 1

Reviewer 1 Report

The introduction is brief, but that is quite normal for health sciences papers. It provides a socio-political and health risk context in which the study is located.

What is missing, however, is a brief literature review of literature in the area of health utilization in refugee camps, forming maybe the final paragraph of the introduction. It states there is 'little', per the abstract - then exactly what is the little that does exist, where are these studies, sample and methods, etc.

The discussion section brings in come comparison of other studies. These studies could form the literature overview up front in the article, so as to 'book end' the study. For example, some comparison of the original research is made with refugees at a refugee camp in Lesbos, Greece. As a reader, I would like to know the Lesbossample and methods to be able to understand the strength of comparisons being made in the discussion later on.

The rest of the manuscript is fine - interesting and valuable to the academic community. Language and structure is good. Tables and diagrams are appropriate.

Author Response

Dear Reviewer, dear Editor,

thank you for your positive and insightful comments and for assisting us in improving the manuscript. Below you will find a point-by-point reply to your comments:

The introduction is brief, but that is quite normal for health sciences papers. It provides a socio-political and health risk context in which the study is located. What is missing, however, is a brief literature review of literature in the area of health utilization in refugee camps, forming maybe the final paragraph of the introduction. It states there is 'little', per the abstract - then exactly what is the little that does exist, where are these studies, sample and methods, etc.

Thank you for your feedback. Although some data on health care utilization in refugee camps during the current crisis exists, specific data dressing the health care needs in children is extremely scarce. We now included a brief respective literature review on health care utilization by refugees with a particular focus on children and adolescents in the introduction and discussion part: “We previously showed that health care utilization is high in freshly arriving immigrants in Western Europe during the current crisis, especially directly upon arrival at a primary refugee reception center 16. Another study on primary medical care in newly arriving immigrants in Greece showed a similarly high frequency of health care seeking behavior, especially due to infectious and dental pathologies, but also because of psychological diagnoses 13. In Turkish regions close to the Syrian border, refugee children were significantly more likely to visit local emergency departments than local children were 17. Another study by Watts et al. showed that refugee children not only displayed a comparably high disease burden when arriving at a country with high economical standard, but that these diseases also led to significant health care demand during the first year after resettlement 18. However, in spite of the high proportion of migrating children and refugees, data on their particular medical needs during the current crisis is scarce.”

The discussion section brings in come comparison of other studies. These studies could form the literature overview up front in the article, so as to 'book end' the study. For example, some comparison of the original research is made with refugees at a refugee camp in Lesbos, Greece. As a reader, I would like to know the Lesbos sample and methods to be able to understand the strength of comparisons being made in the discussion later on.

Thank you for your insight. We now included the Lesbos study in our introduction. We also modified the discussion section, further explaining the methodological approach by Herman et al. and speculating on the reasons for the differences in the observed visit frequencies between the two cohorts: “The healthcare seeking rates in our cohort were considerably higher than those previously described by Hermans et al. for underage refugees at a camp in Lesbos, Greece 13.  In this study, all consecutive patients who visited doctors from the local “Boat Refugee Foundation” that provided onsite primary medical care at the refugee sites Camp Moria and Caritas hotel were analyzed to assess healthcare seeking behavior during the current crisis. The authors found a mean of 0.01 visit per day in refugees at Lesbos as compared to 0.06 in our cohort arriving in Germany. The observed difference may be due to a different setup of medical care provision and different recording strategies in the two studies. However, the healthcare seeking rates within our cohort after week eight (mean visit of 0.008 per day) are somewhat comparable to those observed by Hermans et al..”.

The rest of the manuscript is fine - interesting and valuable to the academic community. Language and structure is good. Tables and diagrams are appropriate.

Thank you for this positive comment. As pediatricians and general physicians, we think our data can help in improving care taking strategies for young refugees and are grateful for your support in getting our manuscript published!

The Authors

Reviewer 2 Report

This manuscript carefully describes visit patterns by age, sex, and country of origin of children and adolescents in a refugee camp in Germany.  While potentially important to improving the health care for future refugee populations, the complete lack of information on the nature of the conditions for which care was sought limits tremendously the usefulness of the study.  If the authors could address the diagnoses/chief complaints etc and connect those to the literature review where  issues such as infectious disease, chronic disease and physical trauma, as well as depression are mentioned, it would be a much more significant paper. 

That being said, there are a few issues with the current data.  First, the authors need to do a better job of describing their methods given that one must assume the population was very fluid.  Were the camp numbers for a given day or given time period-eg 6 months, one year?? How representative was the described population of all children flowing thru this facility? Even though the authors report higher utilization by infants and toddlers, its striking that the percent of children seeking care is lower than the percent in the population and was reported to be not different from adults.  This is not discussed.  Also the discussion about the needs of unaccompanied minor seems over interpreted given this represents only 20 individuals.  I imagine that given no family members present that there would be encouragement of these adolescents to seek help or connect to support that might be very different from the needs of a toddler with a respiratory condition taken to clinic by a family member. 

Author Response

Reviewer 2

Dear Reviewer, dear Editor,

thank you for your positive and helpful comments. Below you will find a point-by-point reply to your comments:

This manuscript carefully describes visit patterns by age, sex, and country of origin of children and adolescents in a refugee camp in Germany.  While potentially important to improving the health care for future refugee populations, the complete lack of information on the nature of the conditions for which care was sought limits tremendously the usefulness of the study.  If the authors could address the diagnoses/chief complaints etc and connect those to the literature review where issues such as infectious disease, chronic disease and physical trauma, as well as depression are mentioned, it would be a much more significant paper. 

Thank you for this suggestion. We agree that a comprehensive analysis of reasons for medical care seeking in our cohort would improve the impact of our work. As such, we now followed your suggestion and have used the last 10 days to conduct a chart review of 100 randomly selected cases from our cohort to analyze the issues that led to medical visits in this representative data. We assessed the ICD-10 diagnosis groups leading to referral and describe the top ten diagnoses in this sample in more detail. Our approach and the results from this analysis are now mentioned in the results section and included as a new graph into the supplementary body of our manuscript. Also, we included a new respective paragraph into the discussion of the paper. The new parts of the manuscript are as follows, in the methods part: “For the analysis of reasons for medical visits as presented in Suppl. Fig. 1, medical records of n=100 random visits of children were taken from the entire dataset employing SPSS, and International Statistical Classification of Diseases and Related Health Problems (ICD-10) coding was used to describe the patient complaints.”. In the results part: “Finally, we conducted a sample analysis to address the reasons for primary health care seeking in our cohort. For this, we took a random sample of n=100 health care visits from the entire dataset and analyzed the complaints on the basis of patiens’ medical record in these children and adolescents (47% male, mean age 6.5 yrs. (range 0-18 years)). In 89% of visits, a complaint was recorded. As shown in Suppl. Fig. 1, the main specific diagnosis groups based on ICD-10 recording included respiratory and infectious diseases (39,44%), followed by diseases of the skin, ear or gastrointestinal tract (Suppl. Fig. 1A). When we furthermore analyzed more specific diagnoses, complaints such as cough, respiratory disease, fever and infections or gastroenteritis occurred as top diagnoses, again illustrating that these acute pathologies appeared to be the most prevalent complaints in our cohort (Suppl. Fig. 1B).”. In the discussion part: “Also the finding of higher number of clinic visits in autumn and winter is in accordance with the high rate of communicable diseases in these seasons 26. Although all housings were heated and all inhabitants of the refugee camp had regular access to sanitation, the crowdedness of the living situation rendered them particularly vulnerable to the fast spreading of communicable diseases, particularly during cold season 8,9,12,13. This is also reflected by the analysis of diagnoses in a sample of our cohort: although we did not conduct a comprehensive analysis of all healthcare visits in our setting, our preliminary calculation shows that the main reasons for visiting the onsite medical team in our cohort were acute complaints such as infection, cough, fever or other potentially infectious diseases. This observation, again, goes in line with that of us and other authors describing the increased infection susceptibility in young refugees and their increased health care demand due to infection related pathologies or other acute complaints 9,27,28.”. Also, we included a brief comment on these analyses in the paragraph on general limitations of our work in the discussion section. And finally, we included the data in the supplementary body: “

Supplementary Fig. 1. Top ten ICD-10 based diagnose groups (A) and diagnoses (B) as identified in a random sample of n=100 healthcare visits within the cohort.”.

That being said, there are a few issues with the current data.  First, the authors need to do a better job of describing their methods given that one must assume the population was very fluid.  Were the camp numbers for a given day or given time period-eg 6 months, one year?? How representative was the described population of all children flowing thru this facility?

We thank the reviewer for this comment and apologize that our methods section was obviously not entire clear to this point. Therefore, we now rephrased this section. The population in this camp was indeed fluid as the study included all children that were admitted to this camp from opening to closing. Because no children were omitted from the analysis, it is fully representative for the entire migrant cohort housed in the reception site. This section now reads: “N=3.104 migrants were housed at a temporary reception site in Celle, Northern Germany from it´s opening on September 4th 2015 to closing on July 4th 2016. This study analyzed all n=1.423 children and adolescents admitted to this emergency refugee shelter. The median duration of camp stay was 34 days. All children and adolescents, that entered the camp from opening to closing and all medical data was included into the analysis.”.

Even though the authors report higher utilization by infants and toddlers, its striking that the percent of children seeking care is lower than the percent in the population and was reported to be not different from adults.  This is not discussed. 

Thank you for raising this valid point. Indeed, the overall health care utilization in children was not higher than that described for a similar cohort of all ages described by us previously (Wetzke et al., IJERPH 2018). This is most probably an effect of the large number of teenagers included who have not utilized health care more often than adults. As they make up for most of the included children, overall utilization was not higher than in adults. We now mention and explain this in a new, brief part in the discussion: “The overall rate of healthcare seeking in children and adolescents was not significantly higher than that that previously reported by us for refugees of all ages in a similar setting, which may be due to the high percentage of adolescents in our current analysis group that did not display an increased demand for medical care as compared to the previously analyzed migrants 16. However, in our cohort, a significant inverse correlation of age and healthcare demand was found.”.

Also the discussion about the needs of unaccompanied minor seems over interpreted given this represents only 20 individuals.  I imagine that given no family members present that there would be encouragement of these adolescents to seek help or connect to support that might be very different from the needs of a toddler with a respiratory condition taken to clinic by a family member. 

Thank you for your suggestion. We fully agree that the low number of unaccompanied individuals is a limitation of our work and tried to make this more clear in the revised manuscript version. Accordingly, we rephrased this section, so it now reads: “As the number of unaccompanied children and adolescent refugees within our cohort is extremely small (n=20), we can only cautiously speculate that the observed differences in healthcare seeking behavior in this subgroup reflect the demands of unaccompanied minors during the current crisis. We observed a higher healthcare seeking rate in unaccompanied as compared to accompanied minors within our cohort.”. Also, the low number of unaccompanied minors in our analyses is mentioned in the paragraph on general limitations of our work.

Thank you again for your comments that, in our view, helped to significantly increase the quality of our manuscript. We are grateful for your support!

The Authors

Round 2

Reviewer 2 Report

The authors have been very responsive to prior reviews.  On additional point might be to add in a brief sentence about how representative the randomly selected sample of chart reviews were of the over all population in terms of age, gender.  If not significantly different just saying so makes the newly added diagnosis findings stronger.

Author Response

Dear reviewer, dear editor,

thank you for your fruitful comments and in assisting us to make the manuscript stronger. 

We now analysed the random visit sample vs. the total visits and found no significant differences in terms of gender, age, region of origin and duration of stay. 

This section now reads:

The random visit sample did not significantly differ from total visits in terms of gender (p=0.104), age (p=0.329), region of origin (p=0.941) and duration of stay (p=.877).

Kind regards,
the authors